# BOSS: Bayesian Optimization over String Spaces

**Henry B. Moss**
STOR-i Centre for Doctoral Training
Lancaster University, UK
h.moss@lancaster.ac.uk

**Daniel Beck**
Computing and Information Systems
University of Melbourne, Australia
d.beck@unimelb.edu.au

**Javier González**
Microsoft Research
Cambridge, UK

**David S. Leslie**
Dept. of Mathematics
and Statistics
Lancaster University, UK

**Paul Rayson**
School of Computing
and Communications
Lancaster University, UK

## Abstract

This article develops a Bayesian optimization (BO) method which acts directly over raw strings, proposing the first uses of string kernels and genetic algorithms within BO loops. Recent applications of BO over strings have been hindered by the need to map inputs into a smooth and unconstrained latent space. Learning this projection is computationally and data-intensive. Our approach instead builds a powerful Gaussian process surrogate model based on string kernels, naturally supporting variable length inputs, and performs efficient acquisition function maximization for spaces with syntactical constraints. Experiments demonstrate considerably improved optimization over existing approaches across a broad range of constraints, including the popular setting where syntax is governed by a context-free grammar.

## 1  Introduction

Many tasks in chemistry, biology and machine learning can be framed as optimization problems over spaces of strings. Examples include the design of synthetic genes [González et al., 2014, Tanaka and Iwata, 2018] and chemical molecules [Griffiths and Hernández-Lobato, 2020, Gómez-Bombarelli et al., 2018], as well as problems in symbolic regression [Kusner et al., 2017] and kernel design [Lu et al., 2018]. Common to these applications is the high cost of evaluating a particular input, for example requiring resource and labor-consuming wet lab tests. Consequently, most standard discrete optimization routines are unsuitable, as they require many evaluations.

Bayesian Optimization [Shahriari et al., 2015, BO] has recently risen as an effective strategy to address the applications above, due to its ability to find good solutions within heavily restricted evaluation budgets. However, the vast majority of BO approaches assume a low dimensional, mostly continuous space; string inputs have to be converted to fixed-size vectors such as bags-of-ngrams or latent representations learned through an unsupervised model, typically a variational autoencoder [Kingma and Welling, 2014, VAE]. In this work, we remove this encoding step and propose a BO architecture that operates directly on raw strings through the lens of convolution kernels [Haussler, 1999]. In particular, we employ a Gaussian Process [Rasmussen, 2003, GP] with a *string kernel* [Lodhi et al., 2002] as the surrogate model for the objective function, measuring the similarity between strings by examining shared non-contiguous sub-sequences. String kernels provide an easy and user-friendly way to deploy BO loops directly over strings, avoiding the expensive architecture tuning required to find a useful VAE. At the same time, by using a kernel trick to work in much richer feature spaces than the bags-of-ngrams vectors, string kernels can encode the non-contiguity known to be informative when modeling genetic sequences [Vert, 2007] and SMILES [Anderson et al., 1987] representations of molecules [Cao et al., 2012](see Figure 1). We show that our string kernel's two

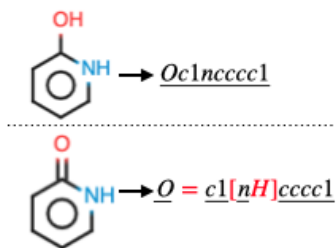

Figure 1: Similar molecules have SMILES strings with local differences (red) but common non-contiguous sub-sequences.

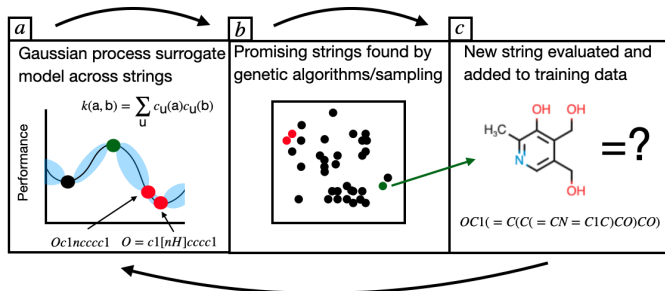

Figure 2: BO loop for molecule design using a string kernel surrogate model (a) and genetic algorithms for acquisition function maximization (b).

parameters can be reliably fine-tuned to model complex objective functions with just a handful of function evaluations, without needing the large collections of unlabeled data required to train VAEs.

Devising a BO framework directly over strings raises the question of how to maximize *acquisition functions*; heuristics used to select new evaluation points. Standard BO uses numerical methods to maximize these functions but these are not applicable when the inputs are discrete structures such as strings. To address this challenge, we employ a suite of genetic algorithms [Whitley, 1994] to provide efficient exploration of string spaces under a range of syntactical constraints.

Our contributions can be summarized as follows:

- We introduce string kernels into BO, providing powerful GP surrogate models of complex objective functions with just two data-driven parameters (Figure 2.a).
- We propose a suite of genetic algorithms suitable for efficiently optimizing acquisition functions under a variety of syntactical constraints (Figure 2.b).
- We demonstrate that our framework out-performs established baselines across four scenarios encompassing a range of applications and diverse set of constraints.

## 2 Related Work

**BO by feature extraction**    BO has previously been applied to find genes with desirable features: a high-cost string optimization problem across a small alphabet of four bases. Genes are represented as either codon frequencies (a bags-of-ngrams of triplets of characters) [González et al., 2014], or as a one-hot-encoding of the genes at each location in the string [Tanaka and Iwata, 2018]. Although these representations are sufficient to allow BO to improve over random gene designs, each mapping discards information known to be important when modeling genes. A bags-of-ngrams representation ignores positional and contextual information by modeling characters to have equal effect regardless of position or context, whereas a one-hot encoding fails to exploit translational invariance. Moreover, by assuming that all potential genes belong to a small fixed set of candidates, González et al. [2014] and Tanaka and Iwata [2018] ignore the need to provide an efficient acquisition optimization routine. This assumption is unrealistic for many real gene design loops and is tackled directly in our work.

**BO with VAEs**    Kusner et al. [2017], Gómez-Bombarelli et al. [2018] and Lu et al. [2018] use VAEs to learn latent representations for string spaces following the syntactical constraints given by context-free grammars (CFG). Projecting a variable-length and constrained string space to an unconstrained latent space of fixed dimensions requires a sophisticated mapping, which in turn requires a lot of data to learn. As BO problems never have enough string-evaluation pairs to learn a supervised mapping, VAEs must be trained to reconstruct a large collection of valid strings sampled from the CFG. A representation learned in this purely unsupervised manner will likely be poorly-aligned with the problem's objective function, under-representing variation and over-emphasizing sub-optimal areas of the original space. Consequently, VAE's often explore only limited regions of the space and have 'dead' areas that decode to invalid strings [Griffiths and Hernández-Lobato, 2020]. Moreover, performance is sensitive to the arbitrary choice of the closed region of latent space considered for BO.

| String, $\mathbf{s}$ | Sub-sequence Occurrence, $\mathbf{u}$ | | | Sub-sequence Contribution, $c_{\mathbf{u}}(\mathbf{s})$ | | |
|---|---|---|---|---|---|---|
| | "genic" | "geno" | "ge" | "genic" | "geno" | "ge" |
| "genetics" | "**gen**et**ic**s" | | "**ge**netics" "**ge**netics" | $\lambda_m^5 \lambda_g^2$ | $0$ | $\lambda_m^2(1+\lambda_g^2)$ |
| "genomic" | "**gen**om**ic**" | "**geno**mic" | "**ge**nomic" | $\lambda_m^5 \lambda_g^2$ | $\lambda_m^4$ | $\lambda_m^2$ |
| "genomes" | | "**geno**mes" | "**ge**nomes" "**ge**nom**e**s" | $0$ | $\lambda_m^4$ | $\lambda_m^2(1+\lambda_g^4)$ |

Table 1: Occurrences (left panel) and respective contributions function values (right panel) of sample sub-sequences when evaluating the strings "genetics", "genomic" and "genomes".

**Evolutionary algorithms in BO**    The closest existing idea to our work is that of Kandasamy et al. [2018], where an evolutionary algorithm optimizes acquisition functions over a space of neural network architectures. However, their approach does not support syntactically constrained spaces and, as it is based solely on local mutations, cannot perform the global search required for large string spaces. Moreover, as their kernel is based on an *optimal transport* distance between individual network layers, it does not model the non-contiguous features supported by string kernels. Contemporaneous work of Swersky et al. [2020] also considers BO over strings and proposes an evolutionary algorithm based on generative modeling for their acquisition function optimization. However, their approach relies on ensembles of neural networks rather than GP surrogate models, is suitable for strings of up to only 100 characters and does not support spaces with syntactic constraints.

## 3   Preliminaries

**Bayesian Optimization**    In its standard form, BO seeks to maximize a smooth function $f : \mathcal{X} \to \mathbb{R}$ over a compact set $\mathcal{X} \subset \mathbb{R}^d$ in as few evaluations as possible. Smoothness is exploited to predict the performance of not yet evaluated points, allowing evaluations to be focused into promising areas of the space. BO loops have two key components - a *surrogate model* and an *acquisition function*.

*Surrogate model* To predict the values of $f$ across $\mathcal{X}$, a surrogate model is fit to the previously collected (and potentially noisy) evaluations $D_t = \{(\mathbf{x}_i, y_i)\}_{i=1,...,t}$, where $y_i = f(\mathbf{x}_i) + \epsilon_i$ for iid Gaussian noise $\epsilon_i \sim \mathcal{N}(0, \sigma^2)$. As is standard in the literature, we use a GP surrogate model [Rasmussen, 2003]. A GP provides non-parametric regression of a particular smoothness controlled by a kernel $k : \mathcal{X} \times \mathcal{X} \to \mathbb{R}$ measuring the similarity between two points.

*Acquisition function* The other crucial ingredient for BO is an acquisition function $\alpha_t : \mathcal{X} \to \mathbb{R}$, measuring the utility of making a new evaluation given the predictions of our surrogate model. We use the simple yet effective search strategy of expected improvement (EI): evaluating points yielding the largest improvement over current evaluations. Although any BO acquisition function is compatible with our framework, we choose EI as it provides an effective search whilst not incurring significant BO overheads. Under a GP, EI has a closed form expression and can be efficiently calculated (see Shahriari et al. [2015]). A single BO loop is completed by evaluating the location with maximal utility $\mathbf{x}_{t+1} = \mathrm{argmax}_{\mathbf{x} \in \mathcal{X}} \, \alpha_t(\mathbf{x})$ and is repeated until the optimization budget is exhausted.

**String Kernels (SKs)**    SKs are a family of kernels that operate on strings, measuring their similarity through the number of *sub-strings* they share. Specific SKs are then formally defined by the particular definition of a sub-string they encompass, which defines the underlying feature space of the kernel. In this work, we employ the *Sub-sequence String Kernel* (SSK) [Lodhi et al., 2002, Cancedda et al., 2003], which uses *sub-sequences* of characters as features. The sub-sequences can be non-contiguous, giving rise to an exponentially-sized feature space. While enumerating such a space would be infeasible, the SSK uses the kernel trick to avoid computation in the primal space, enabled via an efficient dynamic programming algorithm. By matching occurrences of sub-sequences, SSKs can provide a rich contextual model of string data, moving far beyond the capabilities of popular bag-of-ngrams representations where only contiguous occurrences of sub-strings are modeled.

Formally, an $n^{th}$ order SSK between two strings $\mathbf{a}$ and $\mathbf{b}$ is defined as

$$k_n(\mathbf{a}, \mathbf{b}) = \sum_{\mathbf{u} \in \Sigma^n} c_{\mathbf{u}}(\mathbf{a}) c_{\mathbf{u}}(\mathbf{b}) \quad \text{for} \quad c_{\mathbf{u}}(\mathbf{s}) = \lambda_m^{|\mathbf{u}|} \sum_{1 < i_1 < .. < i_{|\mathbf{u}|} < |\mathbf{s}|} \lambda_g^{i_{|\mathbf{u}|} - i_1} \mathbb{1}_{\mathbf{u}}((s_{i_1}, .., s_{i_{|\mathbf{u}|}})),$$

where $\Sigma^n$ denotes the set of all possible ordered collections containing up to $n$ characters from our alphabet $\Sigma$, $\mathbb{1}_{\mathbf{x}}(\mathbf{y})$ is the indicator function checking if the strings $\mathbf{x}$ and $\mathbf{y}$ match, and the match decay $\lambda_m \in [0, 1]$ and gap decay $\lambda_g \in [0, 1]$ are kernel hyper-parameters. Intuitively, $c_{\mathbf{u}}(\mathbf{s})$ measures the contribution of sub-sequence $\mathbf{u}$ to string $\mathbf{s}$, and the choices $\lambda_m$ and $\lambda_g$ control the relative weighting of long and/or highly non-contiguous sub-strings (Table 1). To allow the meaningful comparison of strings of varied lengths, we use a normalized string kernel $\tilde{k}_n(\mathbf{a}, \mathbf{b}) = k_n(\mathbf{a}, \mathbf{b})/\sqrt{k_n(\mathbf{a}, \mathbf{a})k_n(\mathbf{b}, \mathbf{b})}$.

# 4 Bayesian Optimization Directly On Strings

In string optimization tasks, we seek the optimizer $\mathbf{s}^* = \operatorname{argmax}_{\mathbf{s} \in S} f(\mathbf{s})$ of a function $f$ across a set of strings $S$. In this work, we consider different scenarios for $S$ arising from three different types of syntactical constraints and a sampling-based approach for when constraints are not fully known. In Section 5 we demonstrate the efficacy of our proposed framework across all four scenarios.

1. **Unconstrained** Any string made exclusively from characters in the alphabet $\Sigma$ are allowed. $S$ contains all these strings of any (or a fixed) length.
2. **Locally constrained** $S$ is a collection of strings of fixed length, where the set of possible values for each character depends on its position in the string, i.e. the character $s_i$ at location $i$ belongs to the set $\Sigma_i \subseteq \Sigma$.
3. **Grammar constrained** $S$ is the set of strings made from $\Sigma$ that satisfy the syntactical rules specified by a context-free grammar.
4. **Candidate Set**. A space with unknown or very complex syntactical rules, but for which we have access to a large collection $S$ of valid strings.

## 4.1 Surrogate Models for String Spaces

To build a powerful model across string spaces, we propose using an SSK within a GP. However, the vanilla SSK presented above is not immediately suitable due to its substantial computational cost. In contrast to most applications of GPs, BO surrogates are trained on small datasets and so the computational bottleneck is not inversion of Gram matrices. Instead, the primary contributors to cost are the many kernel evaluations required to maximize acquisition functions. Therefore, we develop two modifications to improve the efficiency and scalability of our SSK.

*Efficiency* Using the dynamic program proposed by Lodhi et al. [2002], obtaining a single evaluation of an $n^{th}$ order SSK is $O(nl^2)$, where $l = \max(|\mathbf{a}|, |\mathbf{b}|)$. For our applications where many kernel evaluations are to be made in parallel, we found the vectorized formulation of Beck and Cohn [2017] to be more appropriate. Although, having a larger complexity of $O(nl^3)$, a vectorized formulation can exploit recent advancements in parallel processing and in practice was substantially faster. Moreover, Beck and Cohn [2017]'s formulation provides gradients with respect to the kernel parameters, allowing their fine-grained tuning to a particular optimization task. We found the particular string kernel proposed by Beck and Cohn [2017] — with individual weights for each different sub-sequence length — to be overly flexible for our BO applications. We adapt their recursive algorithm for our SSK (Appendix A).

*Scalability* Even with a vectorized implementation, SSKs are computationally demanding for long strings. Comprehensively tackling the scalability of string kernels is beyond the scope of this work and is an area of future research. However, we perform a simple novel approximation to allow demonstrations of BO for longer sequences: we split sequences into $m$ parts, applying separate string kernels (with tied kernel parameters) to each individual part and summing their values. This reduces the complexity of kernel calculations from $O(nl^3)$ to $O(nl^3/m^2)$ without a noticeable effect on performance (Section 5.2). Moreover, the $m$ partial kernel calculations can be computed in parallel.

## 4.2 Acquisition function optimization over String Spaces

We now present a suite of routines providing efficient acquisition function optimization under different types of syntactical constraints. In particular, we propose using *genetic algorithms* (GA) [Whitley, 1994], biologically inspired optimization routines that successively evaluate and evolve populations of $n$ strings. Candidate strings undergo one of two stochastic perturbations: a *mutation* operation producing a new offspring string from a single parent, and a *crossover* operation combining

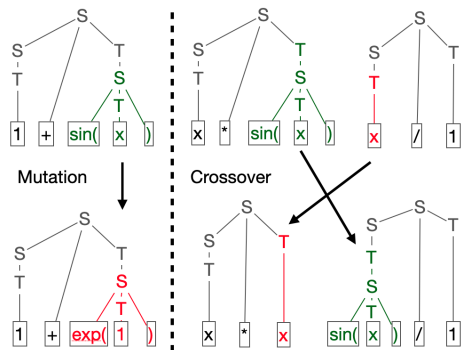

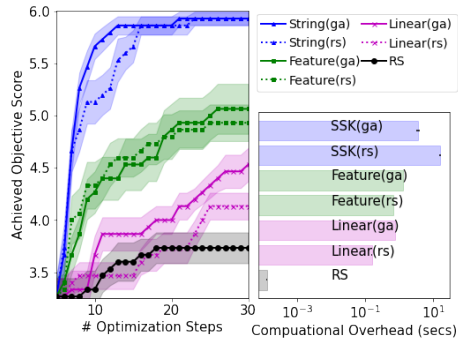

Figure 3: Mutations and crossover of arithmetic expressions following the grammar:
$S \to S$ '+' $T \mid S$ '*' $T \mid S$ '/' $T \mid T$
$T \to$ '$sin($' $S$ '$)$' $\mid$ '$exp($' $S$ '$)$' $\mid$ '$x$' $\mid$ '$1$'.

Figure 4: Performance and computational overhead when searching for binary strings of length 20 with the most non-overlapping occurrences of "101" (higher is better).

attributes of two parents to produce two new offspring. GAs are a natural choice for optimizing acquisition functions as they avoid local maxima by maintaining diversity and the evolution can be carefully constructed to ensure compliance to syntactical rules. We stress that GAs require many function evaluations and so are not suitable for optimizing a high-cost objective function, just for this 'inner-loop' maximization. To highlight robustness, the parameters of our GAs are not tuned to our individual experiments (Appendix F). When syntactical rules are poorly understood and cannot be encoded into the optimization, we recommend the simple but effective strategy of maximizing acquisition functions across a random sample of valid strings.

**GAs for unconstrained and locally constrained string spaces** For our first two types of syntactical constraints, standard definitions of crossover and mutation are sufficient. For mutation, a random position $i$ is chosen and the character at this point is re-sampled uniformly from the set of permitted characters $\Sigma_i$ (or just $\Sigma$) for locally constrained (unconstrained) spaces. For crossover, a random location is chosen within one of the parent strings and the characters up until the crossover point are swapped between the parents. Crucially, the relative positions of characters in the strings are not changed by this operation and so the offspring strings still satisfy the space's constraints.

**GA for grammar-constrained string spaces** Context free grammars (CFG) are collections of rules able to encode many common syntactical constraints (see Appendix B and Hopcroft et al. [2001]). While it is difficult to define character-level mutation and crossover operations that maintain grammatical rules over strings of varying length, suitable operations can be defined over parse trees, structures detailing the grammatical rules used to make a particular string. Following ideas from grammar-based genetic programming [Mckay et al., 2010], mutations randomly replace sub-trees with new trees generated from the same head node, and crossover swaps two sub-trees sharing a head node between two parents (see Figure 3). When sampling strings from the grammar to initialize our GA and perform mutations, the simple strategy of building parse trees by recursively choosing random grammar rules produces long and repetitive sequences. We instead employ a sampling strategy that down-weights the probability of selecting a particular rule based on the number of times it has already occurred in the current parse tree branch (Appendix C).

## 5 Experiments

We now evaluate our proposed BO framework on tasks from a range of fields and syntactical constraints. Our code is available at *github.com/henrymoss/BOSS* and is built upon the Emukit Python package [Paleyes et al., 2019]. All results are based on runs across 15 random seeds, showing the mean and a single standard error of the best objective value found as we increase the optimization budget. The computational overhead of BO (the time spent fitting the GP and maximizing the acquisition function) is presented as average wall-clock times. Although acquisition function calculations could be parallelized across the populations of our GA at each BO step, we use a single-core Intel Xeon 2.30GHz processor to paint a clear picture of computational cost.

| Problem Definition | | | Mean performance with std error (2 s.f.) | | | | |
|---|---|---|---|---|---|---|---|
| Objective | Space | Steps | SSK (ga) | SSK (rs) | Feature (ga) | Linear (ga) | RS |
| # of "101" | $\{0,1\}^{20}$ | 10 | **100 (0.0)** | 96 (1.4) | 97 (2.2) | 58 (3.0) | 58 (2.6) |
| # of "101", no overlaps | $\{0,1\}^{20}$ | 15 | **98 (1.4)** | 94 (2.6) | 76 (4.1) | 64 (2.6) | 60 (3.1) |
| # of "10??1" | $\{0,1\}^{20}$ | 25 | **98 (1.6)** | 95 (1.6) | 64 (2.0) | 64 (3.3) | 56 (2.0) |
| # of "101" in $1^{\text{st}}$ 15 chars | $\{0,1\}^{30}$ | 40 | **91 (2.6)** | 83 (1.7) | 67 (3.0) | 69 (2.6) | 61 (2.3) |
| # of "101" + $\mathcal{N}(0,2)$ | $\{0,1\}^{20}$ | 25 | **98 (2.1)** | 95 (1.4) | 51 (3.9) | 40 (4.0) | 45 (3.8) |
| # of "123" | $\{0,..,3\}^{30}$ | 20 | **81 (2.3)** | 35 (2.8) | 69 (5.4) | 23 (2.0) | 17 (1.5) |
| # of "01??4" | $\{0,..,4\}^{20}$ | 50 | **67 (4.5)** | 38 (2.6) | 35 (4.0) | 33 (3.1) | 29 (2.6) |

Table 2: Optimization of functions counting occurrences of a particular pattern within strings of varying lengths and alphabets ("?" matches any single character). Evaluations are standardized $\in [0, 100]$ and higher scores show superior optimization. Our SSK provides particularly strong performance for complicated patterns (red) or when evaluations are contaminated with Gaussian noise (blue). Our GA acquisition maximizer is especially effective for large alphabets (yellow).

**Considered BO approaches**   For problems with fixed string-length, we compare our SSK with existing approaches to define GP models over strings. In particular, we apply the squared exponential (SE) kernel [Rasmussen, 2003] to a bags-of-ngrams feature representation of the strings. SSKs (feature) representations consider sub-sequences of up to and including five non-contiguous (contiguous) characters, with additional choices demonstrated in Appendix D. We also provide a linear kernel applied to one-hot encodings of each character, a common approach for BO over categorical spaces. The strategy of sequentially querying random strings is included for all plots and we introduce task-specific baselines alongside their results. After a random initialization of $\min(5, |\Sigma|)$ evaluations, kernel parameters are re-estimated to maximize model likelihood before each BO step.

## 5.1   Unconstrained Synthetic String Optimization

We first investigate a set of synthetic string optimization problems over unconstrained string spaces containing all strings of a particular length built from a specific alphabet. Objective functions are then defined around simple tweaks of counting the occurrence of a particular sub-string. Although these tasks seem simple, we show in Appendix D that they are more difficult than the synthetic benchmarks used to evaluate standard BO frameworks. The results for seven synthetic string optimization tasks are included in Table 2, with a deeper analysis of a single task in Figure 4. Additional figures for the remaining tasks showing broadly similar behavior are included in the supplement. To disentangle the benefits provided by the SSK and our GA, we consider two acquisition optimizers: random search across $10,000$ sample strings (denoted *rs* and not to be confused with the random search used to optimize the original objective function) as well as our genetic algorithms (*ga*) limited to $\leq 100$ evolutions of a population of size 100. The genetic algorithm is at most as computationally expensive (in terms of acquisition function evaluations) as the random search optimizer, but in practice is usually far cheaper due to the GA's early-stopping.

Figure 4 demonstrates that our approach provides highly efficient global optimization, dramatically out-performing random search and BO with standard kernels. Interestingly, although the majority of our approach's advantage comes from the SSK, our genetic algorithm also contributes significantly to performance, out-performing the random search acquisition function optimizer in terms of both optimization and computational overhead. Although SSKs incur significant BO overheads, they achieve high-precision optimization after far fewer objective queries, meaning a substantial reduction in overall optimization costs for all but the cheapest objective functions. Table 2 shows that our approach provides superior optimization across a range of tasks designed to test our surrogate model's ability to model contextual, non-contiguous and positional information.

## 5.2   Locally Constrained Protein Optimization

For our second set of examples, we consider the automatic design of genes that strongly exhibit some particular property. We follow the set-up of González et al. [2014], which optimizes across the space of all the genes encoding a particular protein. Proteins are sequences made from 20 amino acids, but redundancy in genetic coding means that individual proteins can be represented by many distinct genes, each with differing biological properties. For this experiment, we seek protein representations

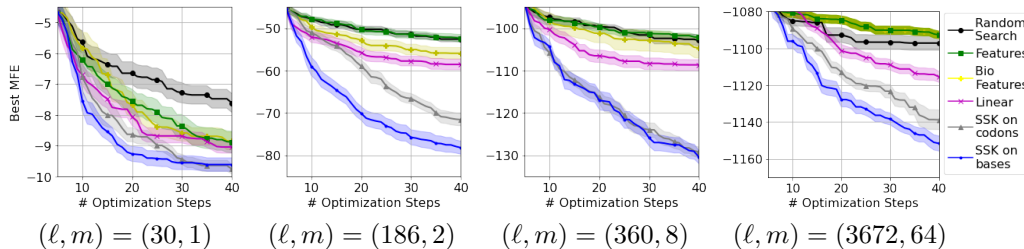

$$(\ell, m) = (30, 1) \qquad (\ell, m) = (186, 2) \qquad (\ell, m) = (360, 8) \qquad (\ell, m) = (3672, 64)$$

Figure 5: Finding the representation with minimal *minimum free-folding energy* (MFE) for proteins of length $\ell$. SSKs are applied to codon or base representations split into $m$ or $3m$ parts, respectively.

with minimal *minimum free-folding energy*, a fundamental biological quantity determined by how a protein 'folds' in 3-D space. The prediction of the most likely free-folding energy for large sequences remains an important open problem [AlQuraishi, 2019], whereas calculating the minimal free-folding energy (across all possible folds) is possible for smaller sequences using the ViennaRNA software [Lorenz et al., 2011]. We acknowledge that this task may not be biologically meaningful on its own, however, as free-folding energy is of critical importance to other down-stream genetic prediction tasks, we believe it to be a reasonable proxy for wet-lab-based genetic design loops. This results in a truly challenging black-box string optimization, requiring modeling of positional and frequency information alongside long-range and non-contiguous relationships.

Each amino acid in a protein sequence can be encoded as one of a small subset of $64$ possible codons, inducing a locally constrained string space of genes, where the set of valid codons depends on the position in the gene (i.e the particular amino acid represented by that position). Alternatively, each codon can be represented as triples of the bases (A,C,T,G), forming another locally constrained string space of three times the length of the codon representation but with a smaller alphabet size of $4$. As well as applying the linear and feature kernels to the base representations, we also consider the domain-specific representation used by González et al. [2014] (denoted as Bio-Features) that counts codon frequencies and four specific biologically inspired base-pairs. Figure 5 demonstrates the power of our framework across $4$ proteins of varying length. Additional details and wall-clock timing are provided in Appendix E. SSKs provide particularly strong optimization for longer proteins, as increasing the length renders the global feature frequencies less informative (with the same representations used for many sequences) and the linear kernel suffers the curse of dimensionality. Note that unlike existing BO frameworks for gene design, our framework explores the large space of all possible genes rather than a fixed small candidate set.

### 5.3    Grammar Constrained String Optimization

We now consider a string optimization problem under CFG constraints. As these spaces contain strings of variable length and have large alphabets, the linear and feature kernel baselines considered earlier cannot be applied. However, we do consider the VAE-based approaches of Kusner et al. [2017] and Gómez-Bombarelli et al. [2018] denoted *GVAE* and *CVAE* for a grammar VAE and character VAE, respectively. We replicate the symbolic regression example of Kusner et al. [2017], using their provided VAEs pre-trained for this exact problem. Here, we seek a valid arithmetic expression that best mimics the relationship between a set of inputs and responses, whilst following the syntactical rules of a CFG (Appendix B). We investigate both BO and random search in the latent space of the VAEs, with points chosen in the latent space decoded back to strings for objective function evaluations (details in Appendix F). We sample $15$ strings for initialization of our GPs, which, for the VAE-approaches, are first encoded to the latent space, before being decoded for evaluation. The invalid strings suggested by *CVAE* are assigned large error.

Figure 6 shows that our approach is able to provide highly efficient BO across a space with complicated syntactical constraints, out-performing the VAE methods which are beaten by even random search (a comparison not made by Kusner et al. [2017]). The difference in starting values for the performance curves in Figure 6 is due to stochasticity when encoding/decoding; initial strings are rarely decoded back to themselves but instead mapped back to a less diverse set. However, sampling directly in the latent space led to a further decrease in initialization diversity. We stress that *CVAE* and *GVAE* were initially designed as models which, using BO-inspired arguments, could generate new valid strings outside of their training data. Consequently, they have previously been tested only in scenarios with

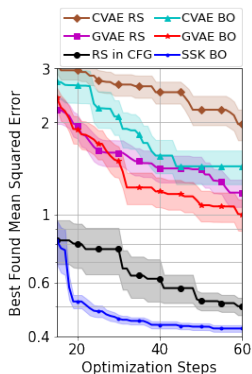

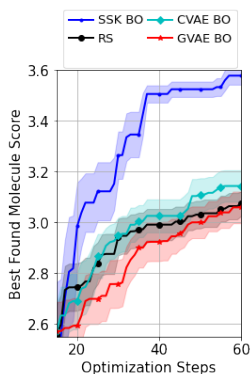

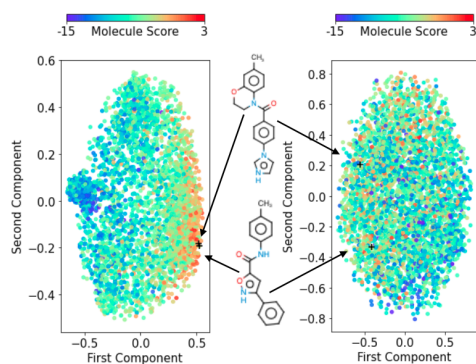

Figure 6: Searching for arithmetic expressions satisfying constraints from a CFG (lower is better).

Figure 7: Searching a candidate set for molecules with desirable properties (higher is better).

Figure 8: Top KPCA components for our SSK (left) and an SE kernel in the *GVAE* (right) for SMILES strings. Our SSK has a smoother internal representation, where 'close' points are structurally similar.

significant evaluation budgets. To our knowledge, we are the first to analyze their performance in the low-resource loops typical in BO.

### 5.4 Optimization Over a Candidate Set

Finally, we return to the task introduced briefly in Figure 2 of searching over SMILES strings to find molecules with desirable properties. As the validity of SMILES strings are governed by complex semantic and syntactic rules that can only be partly explained by a context-free grammar [Kraev, 2018], it is not obvious how to define a GA acquisition function optimizer that can explore the space of all valid molecules. Therefore, we consider an alternative task of seeking high-scoring molecules from within the large collection of $250,000$ candidate molecules used by Kusner et al. [2017] to train a *CVAE* and *GVAE*. Once again, we stress that Kusner et al. [2017]'s primary motivation is to use a large evaluation budget to generate new molecules outside of the candidate set, whereas we consider the simpler but still realistic task of efficiently exploring within the set's limits. At each BO step, we sample 100 candidates, querying those that maximize the acquisition function predicted by our SSK as well as by GPs with SE kernels over the VAE latent spaces. Figure 7 shows that only the SSK allows efficient exploration of the candidate SMILES strings. We hypothesize that the VAEs' poor performance may be partly due to the latent space's dimension which, at $56$, is likely to severely hamper the performance of any BO routine.

**SSK's internal representations** A common way to investigate the efficacy of VAEs is to examine their latent representations. However, even if objective evaluations are smooth across this space [Kusner et al., 2017], this smoothness cannot be exploited by BO unless successfully encapsulated by the surrogate model. Although GPs have no explicit latent space, they have an intrinsic representation that can be similarly examined to provide visualization of a surrogate model's performance. In particular, we apply kernel principal component analysis (KPCA) [Schölkopf et al., 1997] to visualize how SMILES strings map into the feature space. Figure 8 shows the first two KPCA components of our SSK and of an SE kernel within the *GVAE*'s latent space (additional visualizations in Appendix G). Although the latent spaces of the *GVAE* is known to exhibit some smoothness for this SMILES task [Kusner et al., 2017], the smoothness is not captured by the GP model, in contrast with the SSK.

## 6 Discussion

Departing from fixed-length representations of strings revolutionizes the way in which BO is performed over string spaces. In contrast to VAEs, where models are learned from scratch across thousands of parameters, an SSK's structure is predominantly fixed. By hard-coding prior linguistic intuition about the importance of incorporating non-contiguity, our SSKs have just two easily identifiable kernel parameters governing modeling of a particular objective function. We posit that the

additional flexibility of VAEs is not advantageous in BO loops, where there is never enough data to reliably learn flexible models and where calibration is more important than raw predictive strength.

As well as achieving substantially improved optimization, we provide a user-friendly BO building-block that can be naturally inserted into orthogonal developments from the literature, including batch [González et al., 2016], multi-task [Swersky et al., 2013], multi-fidelity [Moss et al., 2020b] and multi-objective [Hernández-Lobato et al., 2016] BO, as well as BO with controllable experimental noise [Moss et al., 2020a] (all with obvious applications within gene and chemical design loops). Moreover, our framework can be extended to other kinds of convolution kernels such as tree [Collins and Duffy, 2002] and graph kernels [Vishwanathan et al., 2010]. This would allow the optimization of other discrete structures that have previously been modeled through VAEs, including networks [Zhang et al., 2019] and molecular graphs [Kajino, 2019].

## Broader Impact

The primary contribution of our work is methodological, providing an efficient and user-friendly framework for optimizing over discrete sequences. As noted in the paper, this is a broad class of problems with a growing interest in the machine learning literature. We hope that our accessible code base will encourage the deployment of our method by practitioners and researchers alike.

We have highlighted two real-world applications by demonstrating efficiency improvements within automatic gene and molecule design loops. Such gains are of considerable interest to biological and chemical research labs. Reducing the wet-lab resources required when searching for chemicals or genes with desirable properties provides not only a substantial environmental and monetary saving, but can even enable new technologies. For example, a fast search over genes is a necessary step in providing custom medical treatments.

On the other hand, wherever our method can be applied to find structures with beneficial properties, it could similarly be used to find structures with malicious properties. Although the optimization itself is automatic, a human should always has the final say in how a particular optimized structure is to be used. This decision making process should in turn incorporate any ethical frameworks specific to the task at hand.

## Acknowledgments

The authors are grateful to reviewers, whose comments and advice have improved this paper. The research was supported by EPSRC, the STOR-i Centre for Doctoral Training and a visiting researcher grant from the University of Melbourne.

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
