[Supplementary Material]

## A Dynamic Programs For SSK Evaluations and Gradients

We now detail recursive calculation strategies for calculating $k_n(\mathbf{a}, \mathbf{b})$ and its gradients with $O(nl^3)$ complexity. A recursive strategy is able to efficiently calculate the contributions of particular substring, pre-calculating contributions of the smaller sub-strings contained within the target string.

Adapting the recursion and notation of Beck and Cohn [2017] to our chosen contribution function, $k_n(\mathbf{a}, \mathbf{b})$ can be calculated by following for $i = 1, ..n - 1$:

$$\mathbf{K}'_0 = \mathbf{1}$$
$$\mathbf{K}'_i = \mathbf{D}^T_{|\mathbf{a}|} \mathbf{K}''_i \mathbf{D}_{|\mathbf{b}|}$$
$$\mathbf{K}''_i = \lambda_m^2 (\mathbf{M} \odot \mathbf{K}'_{i-1})$$
$$k_i = \lambda_m^2 \sum_{j,k} (\mathbf{M} \odot \mathbf{K}'_i)_{j,k},$$

producing the kernel evaluation $k_n(\mathbf{a}, \mathbf{b}) = \sum k_i$. Here, $\odot$ is the Hadamard product, $\mathbf{M}$ is the $|\mathbf{a}| \times |\mathbf{b}|$ matrix of character matches between the two strings ($M_{ij} = \mathbb{1}_{a_i}(b_j)$), and $\mathbf{D}_\ell$ is the $\ell \times \ell$ matrix

$$\mathbf{D}_\ell = \begin{bmatrix} 0 & 1 & \lambda_g & \cdots & \lambda_g^{\ell-2} \\ 0 & 0 & 1 & \cdots & \lambda_g^{\ell-3} \\ \vdots & \vdots & \vdots & \ddots & \vdots \\ 0 & 0 & 0 & \cdots & 1 \\ 0 & 0 & 0 & \cdots & 0 \end{bmatrix}.$$

The gradients of $k_n$ with respect to the kernel parameters $\lambda_m$ and $\lambda_g$ can also be calculated recursively. For the kernel gradients with respect to match decay we calculate

$$\frac{\partial \mathbf{K}'_0}{\partial \lambda_m} = \mathbf{0}$$
$$\frac{\partial \mathbf{K}'_i}{\partial \lambda_m} = \mathbf{D}^T_{|\mathbf{a}|} \frac{\partial \mathbf{K}''_i}{\partial \lambda_m} \mathbf{D}_{|\mathbf{b}|}$$
$$\frac{\partial \mathbf{K}''_i}{\partial \lambda_m} = 2\lambda_m (\mathbf{M} \odot \mathbf{K}'_{i-1}) + \lambda_m^2 \left( \mathbf{M} \odot \frac{\partial \mathbf{K}'_{i-1}}{\partial \lambda_m} \right)$$
$$\frac{\partial k_i}{\partial \lambda_m} = \sum_{j,k} \left[ 2\lambda_m (\mathbf{M} \odot \mathbf{K}'_{ijk}) + \lambda_m^2 \left( \mathbf{M} \odot \frac{\partial \mathbf{K}'_{ijk}}{\partial \lambda_m} \right) \right],$$

producing the gradient $\frac{\partial k_n(\mathbf{a}, \mathbf{b})}{\partial \lambda_m} = \sum \frac{\partial k_i}{\partial \lambda_m}$.

Similarly, kernel gradients with respect to gap decay are calculated by

$$\frac{\partial \mathbf{K}'_0}{\partial \lambda_g} = \mathbf{0}$$
$$\frac{\partial \mathbf{K}'_i}{\partial \lambda_g} = \frac{\partial \mathbf{D}^T_{|\mathbf{a}|}}{\partial \lambda_g} \mathbf{K}''_i \mathbf{D}_{|\mathbf{b}|} + \mathbf{D}^T_{|\mathbf{a}|} \frac{\partial \mathbf{K}''_i}{\partial \lambda_g} \mathbf{D}_{|\mathbf{b}|} + \mathbf{D}^T_{|\mathbf{a}|} \mathbf{K}''_i \frac{\partial \mathbf{D}_{|\mathbf{b}|}}{\partial \lambda_g}$$
$$\frac{\partial \mathbf{K}''_i}{\partial \lambda_g} = \lambda_m^2 \left( \mathbf{M} \odot \frac{\partial \mathbf{K}'_{i-1}}{\partial \lambda_g} \right)$$
$$\frac{\partial k_i}{\partial \lambda_g} = \lambda_m^2 \sum_{j,k} \left( \mathbf{M} \odot \frac{\partial \mathbf{K}'_{ijk}}{\partial \lambda_g} \right),$$

producing the gradient $\frac{\partial k_n(\mathbf{a}, \mathbf{b})}{\partial \lambda_g} = \sum \frac{\partial k_i}{\partial \lambda_g}$, where $\frac{\partial \mathbf{D}_\ell}{\partial \lambda_g}$ is the $\ell \times \ell$ matrix

$$\frac{\partial \mathbf{D}_\ell}{\partial \lambda_g} = \begin{bmatrix} 0 & 0 & 1 & 2\lambda_g & 3\lambda_g^2 & \cdots & (\ell-2)\lambda_g^{\ell-3} \\ 0 & 0 & 0 & 1 & 2\lambda_g & \cdots & (\ell-3)\lambda_g^{\ell-4} \\ 0 & 0 & 0 & 0 & 1 & \cdots & (\ell-4)\lambda_g^{\ell-5} \\ \vdots & \vdots & \vdots & \vdots & \vdots & \ddots & \vdots \\ 0 & 0 & 0 & 0 & 0 & \cdots 1 \\ 0 & 0 & 0 & 0 & 0 & \cdots & 0 \end{bmatrix}.$$

# B  Context-free Grammars

Context-free grammars (CFG) are 4-tuples $G = (V, \Sigma, R, S)$, consisting of:

- a set of non-terminal symbols $V$,
- a set of terminal symbols $\Sigma$ (also known as an alphabet),
- a set of production rules $R$,
- a non-terminal starting symbol $S$ from which all strings are generated.

Production rules are simple maps permitting the swapping of non-terminals with other non-terminals or terminals. All strings generated by the CFG can be broken down into a (non-unique) tree of production rules with the non-terminal starting symbol $S$ at its head. These are known as the parse trees and are demonstrated in Figure 3 in the main paper.

The CFG for the symbolic regression task of Section 5.3 is given by the following rules:

$S \to S$ '+' $T$
$S \to\ S$ '$*$' $T$
$S \to S$ '/' $T$
$S \to\ T$
$T \to$ '(' $S$ ')'
$T \to$ '$sin\,($' $S$ ')'
$T \to$ '$exp\,($' $S$ ')'
$T \to$ '$x$'
$T \to$ '$1$'
$T \to$ '$2$'
$T \to$ '$3$',

where $V = \{S, T\}$ and $\Sigma = \{+, *, /, x, 1, 2, 3\}$. Although each individual production rule is a simple replacement operation, the combination of many such rules can specific a string space with complex syntactical constraints. For example, these 11 rules are able to specify that the string '(sin(2*x)+3(x*(2+exp(x))))+1/2' is valid but that '(sin(2*x)+3(x*(2+exp(x)))+1/2' (with invalid bracket closing) is not.

**Sampling from the CFG**. One of the advantages of CFGs is that it is easy (and cheap) to generate large collections of valid strings by recursively sampling production rules. However, when sampling strings from the grammar, we found this simple sampling strategy to produce long and repetitive strings. For our BO applications, where sample diversity is key, we instead employed a sampling strategy that down-weights the probability of selecting a particular rule based on the number of times it has already occurred in the parse tree. In particular, the probability of applying a particular rule to a non-terminal is proportional to $c^n$, where $n$ is the number of occurrences of that rule in the current branch and $c$ is a discount factor (set to $0.1$ in all our experiments). The construction of this sampler ensures that a wide range of production rules are used when generating from the CFG.

# C  Genetic Algorithms

We now provide implementation details for our GA acquisition function optimizers. During each GA step, populations are refined through stochastic biologically-inspired operations, providing a population achieving (on average) higher scores. The GA begins with a randomly sampled population and ends once the best string in the population stops improving between iterations (Algorithm 1). The

---
**Algorithm 1** Genetic Algorithms for Acquisition Function Maximization
---
1: **function GA**$(p_t, p_c, p_m, N)$
2:    $n \leftarrow 0$
3:    Sample $N$ strings for initial population $P_0$
4:    Evaluate acquisition function $A_0 \leftarrow \alpha(P_0)$
5:    Store current best value $\alpha_{best} \leftarrow \max(A_0)$
6:    **while** $\alpha_{best} = \max(A_n)$ **do**
7:      Begin new iteration $n \leftarrow n + 1$
8:      Evolve population $P_n \leftarrow$ **EVOLVE**$(P_{n-1}, p_t, p_c, p_m)$
9:      Evaluate acquisition function $A_n \leftarrow \alpha(P_n)$
10:     Store current best value $\alpha_{best} \leftarrow \max(\max(A_{n-1}), \alpha_{best})$
11:   **return** String achieving score $\alpha_{best}$
---

---
**Algorithm 2** Evolution of Genetic Algorithm Populations
---
1: **function EVOLVE**$(P, p_t, p_c, p_m)$
2:    Initialize new population $P_{new} \leftarrow \emptyset$
3:    **while** $|P_{new}| < |P|$ **do**
4:      Collect a candidate string $s_1 \leftarrow$ **TOURNAMENT**$(P, p_t)$
5:      Sample $r \sim U[0, 1]$
6:      **if** $r < p_c$ **then**
7:        Sample another candidate string $s_2 \leftarrow$ **TOURNAMENT**$(P, p_t)$
8:        Perform crossover $s_1, s_2 \leftarrow$ **CROSSOVER**$(s_1, s_2)$
9:        Sample $r_1, r_2 \sim U[0, 1]$
10:       **if** $r_1 < p_m$ **then**
11:         Perform mutation $s_1 \leftarrow$ **MUTATION**$(s_1)$
12:       **if** $r_2 < p_m$ **then**
13:         Perform mutation $s_2 \leftarrow$ **MUTATION**$(s_2)$
14:       Add two strings to new population $P_{new} \leftarrow P_{new} \bigcup \{s_1, s_2\}$
15:      **else**
16:        Sample $r \sim U[0, 1]$
17:        **if** $r_1 < p_m$ **then**
18:         Perform mutation $s_1 \leftarrow$ **MUTATION**$(s_1)$
19:       Add string to new population $P_{new} \leftarrow P_{new} \bigcup \{s_1\}$
20:   **return** New population $P_{new}$
---

$N$ strings of the $i + 1^{th}$ population are perturbations of the $i^{th}$ population. To evolve a population (Algorithm 2), a *tournament* process first selects $n$ candidate strings (with replacement) attaining the highest evaluations across random sub-samples of a proportion $p_t$ of the current population. To create the next population, these candidate strings undergo stochastic perturbations: a *mutation* operation producing a new offspring string from a single parent, and a *crossover* operation combining attributes of two parent strings to produce two new offspring. These operations occur with probability $p_c$ and $p_m$ respectively, which, alongside $p_t$, control the level of diversity maintained across populations. To highlight the robustness of our genetic algorithm acquisition optimizer, we do not tune the evolution parameters to each task, using populations of 100 candidate strings and $(p_t, p_c, p_m) = (0.5, 0.75, 0.1)$ for all our experiments. The exact crossover and mutation operators chosen to traverse string spaces under different syntactical constraints are discussed in the main paper.

Figure 9: Comparing random search across standard BO benchmarks (faint) and our synthetic string experiments (bold). For the string tasks, the legend $ALS$ denoted the task with an alphabet of size $A$, strings of length $L$ and counting the occurrences of the pattern $S$.

Figure 10: Optimizing the number of non-overlapping occurrences of "101" in a string of length 20 and alphabet ["0","1"]

# D   Synthetic String Optimization Experiments

Although seemingly simple tasks, our synthetic string optimization tasks of Section 5.1 are deceptively challenging, as only a very small proportion of valid strings produce high scores. In fact, these tasks are considerably more challenging than the common benchmarks used to test standard BO frameworks. Figure 9, shows the performance attained by random search over our synthetic string tasks and standard benchmarks [1]. All objective functions are standardized ($\in [0, 1]$) and we run 1000 optimization steps, plotting the mean and standard error across 25 replications. We see that our easiest synthetic string optimization tasks are among the hardest of the standard benchmark problems to solve with random search, and we expect this to hold similarly for BO.

We now provide comprehensive experimental results across the synthetic string optimization tasks. In Figures 10,11,12,13,14,15 and 16, we show the performance and computational overhead of our string kernels, extending the analysis from the main paper to include a variety of sub-sequence lengths considered by the string and feature-based kernels. We see that the string kernels always provide superior optimization over existing kernels, with the string kernel based on sub-sequences of maximum length 5 consistently among the best. The string kernel is particularly effective for the most complicated objective functions (Figures 11 and 15) and when observations are contaminated by observation noise (Figure 14). For problems with larger alphabets (and so significantly larger search spaces), our genetic algorithm acquisition optimizer dramatically outperforms a larger budget random search optimizer (Figure 13 and 15).

Figure 11: Optimizing the number of occurrences of "10??1" in a string of length 20 and alphabet ["0","1"]

Figure 12: Optimizing the number of occurrences of "101" in the first half of a string of length 30 and alphabet ["0","1"].

Figure 13: Optimizing the number of occurrences of "123" of a string with length 30 and an alphabet of ["0","1","2","3"].

Figure 14: Optimizing the number of occurrences of "101" with observations contaminated by Gaussian noise (with a variance of 2) of a binary string of length 20.

Figure 15: Optimizing the number of occurrences of "01??4" in a string of length 20 and alphabet ["0","1","2,"3","4"]

Figure 16: Optimizing the number of occurrences of "101" in a string of length 20 and alphabet ["0","1"]

# E  Protein Optimization

We now provide additional details for our four protein optimization experiments, each targeting one of the following proteins.

1. Cystic fibrosis transmembrane conductance regulator:

   ```
   TIKENIFGVS.
   ```

2. Invertebrate iridescent virus 6 (IIV-6) (Chilo iridescent virus):

   ```
   MTSRGHLRRAPCCYAFKSATSHQRTRTSLCLASPPAPHCLLLYSHRCLTYFTVDYELSFFCL.
   ```

3. Anaphase-promoting complex subunit 15B:

   ```
   MSTLFPSLLPQVTDSLWFNLDRPCVDENELQQQEQQHQAWLLSIAEKDSSLVPIGKPASEPY
   DEEEEEDDEDDEDSEEDSEDDEDMQDMDEMNDYNESPDDGEIEADMEGAEQDQDQWMI.
   ```

4. Tyrosine-protein kinase abl-1:

   ```
   MGHSHSTGKEINDNELFTCEDPVFDQPVASPKSEISSKLAEEIERSKSPLILEVSPRTPDSV
   QMFRPTFDTFRPPNSDSSTFRGSQSREDLVACSSMNSVNNVHDMNTVSSSSSSSAPLFVALY
   DFHGVGEEQLSLRKGDQVRILGYNKNNEWCEARLYSTRKNDASNQRRLGEIGWVPSNFIAPY
   NSLDKYTWYHGKISRSDSEAILGSGITGSFLVRESETSIGQYTISVRHDGRVFHYRINVDNT
   EKMFITQEVKFRTLGELVHHHSVHADGLICLLMYPASKKDKGRGLFSLSPNAPDEWELDRSE
   IIMHNKLGGGQYGDVYEGYWKRHDCTIAVKALKEDAMPLHEFLAEAAIMKDLHHKNLVRLLG
   VCTHEAPFYIITEFMCNGNLLEYLRRTDKSLLPPIILVQMASQIASGMSYLEARHFIHRDLA
   ARNCLVSEHNIVKIADFGLARFMKEDTYTAHAGAKFPIKWTAPEGLAFNTFSSKSDVWAFGV
   LLWEIATYGMAPYPGVELSNVYGLLENGFRMDGPQGCPPSVYRLMLQCWNWSPSDRPRFRDI
   HFNLENLISSNSLNDEVQKQLKKNNDKKLESDKRRSNVRERSDSKSRHSSHHDRDRDRESLH
   SRNSNPEIPNRSFIRTDDSVSFFNPSTTSKVTSFRAQGPPFPPPPQQNTKPKLLKSVLNSNA
   RHASEEFERNEQDDVVPLAEKNVRKAVTRLGGTMPKGQRIDAYLDSMRRVDSWKESTDADNE
   GAGSSSLSRTVSNDSLDTLPLPDSMNSSTYVKMHPASGENVFLRQIRSKLKKRSETPELDHI
   DSDTADETTKSEKSPFGSLNKSSIKYPIKNAPEFSENHSRVSPVPVPPSRNASVSVRPDSKA
   EDSSDETTKDVGMWGPKHAVTRKIEIVKNDSYPNVEGELKAKIRNLRHVPKEESNTSSQEDL
   PLDATDNTNDSIIVIPRDEKAKVRQLVTQKVSPLQHHRPFSLQCPNNSTSSAISHSEHADSS
   ETSSLSGVYEERMKPELPRKRSNGDTKVVPVTWIINGEKEPNGMARTKSLRDITSKFEQLGT
   ASTIESKIEEAVPYREHALEKKGTSKRFSMLEGSNELKHVVPPRKNRNQDESGSIDEEPVSK
   DMIVSLLKVIQKEFVNLFNLASSEITDEKLQQFVIMADNVQKLHSTCSVYAEQISPHSKFRF
   KELLSQLEIYNRQIKFSHNPRAKPVDDKLKMAFQDCFDQIMRLVDR.
   ```

As each amino acid in these protein sequences can be represented as one of a set of possible codons (triples of bases), the string spaces for these problems are incredibly large, with each space containing $5.53e+4$, $9.48e+33$, $4.81e+49$ and $1.22e+614$ unique strings, respectively. The permitted mappings from amino acids to valid codons are as follows:

$F \rightarrow$ *ttt | ttc*
$L \rightarrow$ *tta | ttg | ctt | ctc | cta, ctg*
$S \rightarrow$ *tct | tcc | tca | tcg | agt | agc*
$Y \rightarrow$ *tat | tac*
$C \rightarrow$ *tgt | tgc*
$W \rightarrow$ *tgg*
$P \rightarrow$ *cct | ccc | cca | ccg*
$H \rightarrow$ *cat | cac*
$Q \rightarrow$ *caa | cag*
$R \rightarrow$ *cgt | cgc | cga | cgg | aga | agg*
$I \rightarrow$ *att | atc | ata*
$M \rightarrow$ *atg*
$T \rightarrow$ *act | acc | aca | acg*
$N \rightarrow$ *aat | aac*
$K \rightarrow$ *aaa | aag*
$V \rightarrow$ *gtt | gtc | gta | gtg*
$A \rightarrow$ *gct | gcc | gca | gcg*
$D \rightarrow$ *gat | gac*
$E \rightarrow$ *gaa | gag*
$G \rightarrow$ *ggt | ggc | gga | ggg.*

Figure 17 extends the analysis of our protein optimization tasks to include the computational overheads incurred by each each BO routine (as measured on a single processor). The high evaluation costs of our SSK means that its overhead is substantially greater than the other approaches. However, in

Figure 17: Optimization performance and computational overhead when finding the representation with minimal *minimum free-folding energy* (MFE) of a protein of length $\ell$. SSKs are applied to codon or base representations split into $m$ or $3m$ parts, respectively.

real gene design loops, this additional computational cost (hours) is negligible compared to the cost and time saved in wet-lab experiments (days). Moreover, the acquisition function calculations can be trivially parallelized across up to 100 cores (the size of the populations used in the GA acquisition function optimizer) as well as across the $m$ partial SSK calculations. If GPUs are available, these can also be used to efficiently calculate SSKs [Beck and Cohn, 2017].

# F   BO in a VAE's Latent Space

To perform BO in the latent space of a VAE, we follow the set-up of Kusner et al. [2017], fitting a GP with an SE kernel and using a multi-start gradient descent acquisition function optimizer. We tried SE kernels with both individual and tied length scales across latent dimensions, however, this did not have a significant effect on performance, possibly due to difficulties in estimating many kernel parameters in these low-data BO problems. In order to perform BO, a compact area of the latent space must be chosen for the search space. Unfortunately, Kusner et al. [2017] do not provide details about how this should be determined. We chose the space containing the most central 75% of representations from the set of strings used to train the VAE (100, 000 arithmetic expressions). We also tried using the space containing all representations from the training data, however, this led to a drop in optimization performance, likely due to less reliable encoding/decoding learned by the VAE in these more sparsely supported parts of the latent space.

(a) SSK on raw SMILES strings.

(b) SE kernel in the *CVAE* latent space.

(c) SE kernel in the *GVAE* latent space.

(d) SSK with poor choices of kernel parameters.

Figure 18: Top two KPCA components visualizing the intrinsic representations of the surrogate models used to predict molecule scores from SMILES strings. Aside from (d), kernel parameters are tuned to maximize GP likelihood over 10 evaluated molecules.

# G  Visualizing BO Surrogate Models

In Section 5.4, we present a kernel principal component analysis (KPCA) visualization of the feature space induced by our SSK. We now extend this analysis to include the VAE competitors. In particular, we perform KPCA on the SE kernel used to define a surrogate model over each VAE's latent representations (Figure 18). All figures show the representations of the same sampled $4,000$ SMILES strings, color-coded to represent their molecule scores (a linear combination of their water-octanol partition coefficient, ring-size and synthetic accessibility). We see that the GP with an SSK produces a significantly smoother KPCA space that the GPs fit in VAE latent space, with the *CVAE* showing slightly more structure than the *GVAE*. This ranking matches the relative performance of the BO routines based on these surrogate models (Figure 7). So although the latent spaces of these VAE have been shown to exhibit some smoothness [Kusner et al., 2017], this is not captured by the GP model. Figure 7.d visualizes the intrinsic representation of an SSK when kernel parameters are purposely chosen to provide a bad fit. We choose very low $\lambda_m$ and high $\lambda_g$ to heavily penalize the long contiguous sub-sequences we know to be informative for this task. The stark difference in smoothness between the visualizations of the tuned and badly-tuned SSKs demonstrates their flexibility as well as the importance of using a representation supervised to the the specific objective function of interest.

## Footnotes

[1] *https://www.sfu.ca/ ssurjano/index.html*