[Reviews · NeurIPS 2020]

Review 1

Summary and Contributions: The authors propose the use of string kernels and genetic algorithms within Bayesian optimization (BO) loops by extending standard BO methods to act directly over raw strings. This is a novel approach and the two technical contributions are: 1) to build a Gaussian process surrogate model based on string kernels, naturally supporting variable length inputs 2) to perform efficient acquisition function maximization for spaces with syntactical constraints by using a genetic algorithm The performance of the method is demonstrated through a set of experiments which show improved optimization over alternative methods across a broad range of constraints.

Strengths: This work is significant and well-motivated by the limitation of existing approaches, where BO over strings is typically performed by mapping inputs into a smooth and unconstrained latent space (which is computationally and data-intensive). It is interesting that unlikely alternative BO approaches for gene design, this framework explores the whole space of possible genes rather than a small candidate set. This seems to be a very desirable feature. The topic is relevant to the NeurIPS community.

Weaknesses: .

Correctness: The claims and methods are well sounded and the previous work is well referenced.

Clarity: This paper is well written and pleasant to read. I believe that clarity is particularly important for papers focused on the intersection of two filed (as is the case for this paper, in which the applications lie in the fields of computational and molecular biology). Minor typo l 78: from --> form

Relation to Prior Work: Previous work is well referenced, both on the technical and application side. This work brings existing techniques (string kernels and genetic algorithms) to BO in a novel way. The authors provide a better solution to an existing problem and improve the performance of BO on it.

Reproducibility: Yes

Additional Feedback: [After rebuttal comment] I would like to thank the authors for the comprehensive response. I will stick to my high score for this paper.


Review 2

Summary and Contributions: Post-rebuttal: This is really a good paper and actually better than average. Therefore I've increased my score. I'd like to see this work published. This submission is concerned with Bayesian optimization on string spaces. The authors use a Gaussian process surrogate model with a sub-sequence string kernel (not novel). Expected improvement is used as acquisition function and optimized with genetic algorithms. Main contribution of this submission is a useful overview and combination of techniques for the addressed problem.

Strengths: Empirical Evaluation ==================== The experiments appear to be comparatively thorough. Particularly, I like that the authors investigate separately the contribution of their string kernel and the optimization via genetic algorithms, and they repeat experiments from the related literature. The authors test their approach in four scenarios with different restrictions on the admissibility of a candidate string: finite but large candidate set, strings constrained via a grammar, strings with differing constraints in each location and no constrains. Novelty ======= This submission does not propose a genuinely novel idea, but combines a number of old and new ideas: the string kernel, a dynamic programming approach to evaluate the latter, genetic algorithms to optimize the acquisition function. In combination with the thorough empirical investigation and the section on related work, this submission is a useful overview on the optimization of problems with string inputs.

Weaknesses: This submission does not propose a per se novel idea. But I do not actually see this as a weakness or limitation.

Correctness: Nothing to complain

Clarity: Nothing to complain

Relation to Prior Work: The authors provide a decent half-a-page overview on related work. I am satisfied.

Reproducibility: Yes

Additional Feedback: Nitpicks ======== l. 109: The letters m and g attached to lambda confused me for a bit--I assumed they were integer variables whose range I could not find out. Maybe you could add a clarification. l. 171-182: The text refers to Appendix C for the details and Appendix C refers to the main text. Please put a complete description of the procedure in one location (preferably Appendix C). l. 265: assigned instead of allocated? Codon is used before it is introduced.


Review 3

Summary and Contributions: In this paper, a novel framework of applying Bayesian optimization directly to the raw strings is proposed. Their work brings a fresh view and solution to the studies of string space. And this work will be welcomed by the community.

Strengths: Explicit statements, solid analysis, and trustful experiments.

Weaknesses: One concern is the GA part. The convergence of GA in string space is not discussed. Efficiency of the GA is addressed, but not explained enough. Relationship with different length of the string is expected.

Correctness: Yes, the claims and method are correct

Clarity: Pretty well written work

Relation to Prior Work: Related work is well discussed in section2, and I think the authors have done a good job in providing introductions and comparisons of these work. But few prior work about string process with other categories of methods is given

Reproducibility: Yes

Additional Feedback: (1) I am curious about the differences between the proposed work and this one: Amortized Bayesian Optimization over Discrete Spaces, by Yulia Rubanov, etc. It seems the general idea is quite similar, for example, they also employ the evolutionary algorithms in BO. (2) What's the maximized length of the string did you test in the experiment? Besides the GA, I am curious if the authors have tried some other methods, such as random forest. (3) The proposed work reminds me of this one: Gaussian processes for natural language processing. In your scenario, have you ever considered using different kernels to build the objective function? (3) I like the discussion of efficiency, but do you have any numerical results? ######################################## The authors have well responsed my questions and concerns. Obviously, this work will be welcomed in the community. By considering the original good work and the satisfied feedback, I'd like to rise my score to "A good submission, accept". Thanks.


Review 4

Summary and Contributions: [re author response: thanks for elaborating---I'll happily stick with my high score.] --- Trying to explore sets of strings, for example those describing molecules, and in particular searching for a single best candidate over them when 1) evaluation is costly and 2) there can be grammatical constraints on the strings is an important problem in computational biology and chemistry. Previous work has either featurized strings inappropriately to use standard Bayesian Optimization or tried to use dense autoencoders (GVAE etc.). This paper argues that autoencoders are to inefficient when you really don’t have much data to pretrain on and that Bayesian Optimization is indeed the way to go: they propose using a string kernel, in particular one that counts (possibly discontiguous) subsequences in a Gaussian Process and verify that indeed this combination wins against previous approaches in a number of different scenarios.

Strengths: * Super easy read. Everything is “obvious” in that it logically follows from what was said before---”obvious” in the best sense. A joy. Nice appendix, too. * Four clearly different but all interesting scenarios to evaluate on. * Sensible domain knowledge (as in grammar constraints really) used for the genetic operations and kernel definition.

Weaknesses: * The effects of m, \lambda_m (which I think is redundant, see below), and \lambda_g are not analyzed in isolation---it would be interesting to see just how sensitive to them the method is. Figure 5 for example shows the effect of m, but mixed up with the data statistic l... * It would have been interesting to see whether this method works as well with other gradient-free optimization methods or whether the genetic approach and in particular the chosen ways of mutating and crossing over matter.

Correctness: Maybe I am missing something, but the way \tilde{k}_n is defined, does \lambda_m matter at all? Does it not cancel out? In that case the method would reduce to a single parameter, which would be nice, but also maybe further improvements could be made by adding a parameter that isn’t “dead”?

Clarity: As mentioned above, this paper is exceptionally well written and an absolute joy to read---everything makes sense. (With an exception for \lambda_m, see above.)

Relation to Prior Work: I am not deeply familiar with BO besides knowing some GP basics, but I had read the GVAE and SD-VAE (Dai et al., 2018, “Syntax-Directed Variational Autoencoder for Structured Data”) papers at the time, so I am familiar with those---the latter could be mentioned and evaluated against, but that is no dealbreaker for me.

Reproducibility: Yes

Additional Feedback: * L75/76 isn’t really clear to me, perhaps that comparison could be fleshed out. * L89: is there a citation for EI or is it fundamental enough (might still want to give a definition)? * L95/96: do *all* string kernels by definition measure similarity through counts of shared substrings? Even the one you propose violates this simplistic definition, no? * L102: indicating the complexity here, if only in a footnote, might be nice. * Figure 4, I would’ve liked to see wallclock time (of only the optimization steps) on the x axis---might be a nice plot for the appendix. * Are the mutation and cross-over positions all equally likely? Maybe some different form of weighting might work better, even if it would incur another hyperparameter to learn. * L200: do they change much throughout the evaluation? This sentence also took me a minute to understand, it might be helpful to elaborate a bit. * Figure 8 might be more readable with smaller and more transparent dots.

[Author Response · NeurIPS 2020]

We thank all reviewers for the time spent on their comprehensive reviews and we are happy that they enjoyed reading our paper. We are particularly glad to see reviewers unanimously agreeing that our work is a well-motivated and significant contribution to the NeurIPS community, providing a needed solution for a problem with immediate applications across a wide range of fields. In this rebuttal, we respond to each reviewer's minor comments individually, as there are no comments shared across reviews.

**Reviewer 1**

Thank you for the extremely positive review. We will fix the one typo reported.

**Reviewer 2**

Thank you for the strongly positive review, arguing for acceptance based on strong performance across our comprehensive and thorough experimental section. In light of your comments around our definitions of $\lambda_m$ and $\lambda_g$, we will endeavour to improve clarity in the final version, emphasising that $\lambda_m$ and $\lambda_g$ are scalar quantities representing our kernel's two hyper-parameters. These parameters tune the contribution of highly non-contiguous sub-sequences in our kernel calculations. We will also address your comments about combining the description of our proposed framework into a single location.

**Reviewer 3**

Thank you for the predominately positive review, stressing the novelty and importance of our work to the NeurIPS community. We regret that minor clarity issues around our use of genetic algorithms prevented you from having as pleasant reading experience as our other reviewers.

No general guarantees exist for the convergence of genetic algorithms over string spaces and their derivation would be a substantial contribution in its own right. However, genetic algorithms were designed with string optimisation problems in mind and are widely used to explore string spaces (see Whitley et al. [1994]). Note that exact convergence of our genetic algorithm is not a requirement for effective BO as exact acquisition maximisation is rarely achieved even when performing BO over continuous spaces. All that is typically required is the identification of a string that scores reasonably highly according to our acquisition function.

We respond to your numbered comments as follows:

1. Rubanova et al. [2020] also consider Bayesian optimisation over strings. However, their approach, using ensembles of neural networks and generative models for acquisition function maximisation, has a smaller scope than our work, seeming suitable for only shorter strings of up to 100 characters and unconstrained string spaces. We will discuss this contemporaneous work (published at the start of this month) in our final version.

2. In Figure 5, we consider a gene optimisation problem over very long strings of 3672 base pairs and we use our locally-constrained genetic algorithm to explore this space of strings. It is unclear how a random forest (typically a classification or regression model) could be used to provide optimisation over discrete (never-mind syntactically constrained) spaces and this is beyond the scope of our work.

3. There is a stream of work combining convolution kernels and GPs for NLP applications (such as the cited Beck and Cohn [2017]), including kernels for modelling trees and graphs. Our method could definitely be extended to such kernels, providing efficient optimisation over additional types of discrete structures. As we mention in our Discussion section, we plan to investigate these avenues in future work.

4. We have presented numerical results for the efficiency of ours (and competitors) BO routines in Figure 4 of the main paper and in Figures 11-17 of our Appendix. We will better signpost this in the final version.

**Reviewer 4**

Thank you for the extremely positive review and helpful minor comments that we will address to improve clarity in the final manuscript. We now deal with your concern that our kernel parameter $\lambda_m$ is redundant after kernel normalisation. As our kernel (defined in line 105) contains the sum over all sub-sequences containing up to $n$ characters, each kernel evaluation is a linear combination of $\lambda_m$ raised to each power in $[2, .., 2n]$. As the coefficients of this linear combination depends on the two considered strings, a dependence on $\lambda_m$ remains even after the normalisation of line 111. The choice of both $\lambda_m$ and $\lambda_g$ is important to the efficacy of the kernel and this is explicitly demonstrated for molecule prediction in Figure 18. In the final version, this figure will be brought into the main paper alongside further clarification of the role of the kernel parameters.

[Meta-Review · NeurIPS 2020]

Reviewers were all very happy with this paper, which represents a principled and practical step forward in performing Bayesian optimization over strings. In the final version of the paper, the authors are encouraged to discuss a recent related paper on Bayesian optimization over discrete spaces, for the benefit of the readership: http://www.auai.org/uai2020/proceedings/329_main_paper.pdf Congratulations on the nice work!